# Novel Therapeutics for Treating Sleep Disorders: New Perspectives on *Maydis stigma*

**DOI:** 10.3390/ijms232314612

**Published:** 2022-11-23

**Authors:** Ryeong-Eun Kim, Darine Froy Mabunga, Hee Jin Kim, Seol-Heui Han, Hahn Young Kim, Chan Young Shin, Kyoung Ja Kwon

**Affiliations:** 1Department of Neuroscience, School of Medicine, Konkuk University, 120 Neungdong-ro, Gwangjin-gu, Seoul 05029, Republic of Korea; 2Department of Pharmacy, Uimyung Research Institute for Neuroscience, Sahmyook University, 815 Hwarangro, Nowon-gu, Seoul 01795, Republic of Korea; 3Department of Neurology, Konkuk Hospital Medical Center, 120-1 Neungdong-ro, Gwangjin-gu, Seoul 05030, Republic of Korea; 4Department of Pharmacology, School of Medicine, Konkuk University, 120 Neungdong-ro, Gwangjin-gu, Seoul 05029, Republic of Korea

**Keywords:** sleep, melatonin receptors, *Maydis stigma*, nutraceuticals, new therapeutics

## Abstract

Sleep is a restorative period that plays a crucial role in the physiological functioning of the body, including that of the immune system, memory processing, and cognition. Sleep disturbances can be caused by various physical, mental, and social problems. Recently, there has been growing interest in sleep. *Maydis stigma* (MS, corn silk) is a female maize flower that is traditionally used as a medicinal plant to treat many diseases, including hypertension, edema, and diabetes. It is also used as a functional food in tea and other supplements. β-Sitosterol (BS) is a phytosterol and a natural micronutrient in higher plants, and it has a similar structure to cholesterol. It is a major component of MS and has anti-inflammatory, antidepressive, and sedative effects. However, the potential effects of MS on sleep regulation remain unclear. Here, we investigated the effects of MS on sleep in mice. The effects of MS on sleep induction were determined using pentobarbital-induced sleep and caffeine-induced sleep disruption mouse models. MS extracts decreased sleep latency and increased sleep duration in both the pentobarbital-induced sleep induction and caffeine-induced sleep disruption models compared to the positive control, valerian root extract. The butanol fraction of MS extracts decreased sleep latency time and increased sleep duration. In addition, β-sitosterol enhances sleep latency and sleep duration. Both MS extract and β-sitosterol increased alpha activity in the EEG analysis. We measured the mRNA expression of melatonin receptors 1 and 2 (MT1/2) using qRT-PCR. The mRNA expression of melatonin receptors 1 and 2 was increased by MS extract and β-sitosterol treatment in rat primary cultured neurons and the brain. In addition, MS extract increased the expression of clock genes including *per1/2*, *cry1/2*, and *Bmal1* in the brain. MS extract and β-sitosterol increased the phosphorylation of ERK1/2 and αCaMKII. Our results demonstrate for the first time that MS has a sleep-promoting effect via melatonin receptor expression, which may provide new scientific evidence for its use as a potential therapeutic agent for the treatment and prevention of sleep disturbance.

## 1. Introduction

Sleep is essential for emotional, physical, and cognitive functions. Sleep has restorative effects after activity, allowing optimal functioning [1,2]. Sleep is responsible for maintaining brain homeostasis, and sleep disturbances including sleep deprivation and loss are closely associated with cognitive impairment, systemic inflammation, and neurodegeneration [3,4]. Therefore, sleep disturbances are potent risk factors and common symptoms for various neurological disorders, including neurovascular diseases, depression, attention deficit hyperactivity disorder (ADHD), autism spectrum disorder (ASD), and neurodegenerative diseases [5,6,7,8].

Major sleep disorders include insomnia, narcolepsy, restless leg syndrome (RLS), and sleep apnea. Insomnia, the most common sleep disorder, is characterized by the inability to fall asleep and stay asleep, resulting in sleep loss and poor sleep quality [9]. Various studies have shown the worldwide prevalence of insomnia at 10–30%, some showing a prevalence even as high as 50–60% [10]. The consequences of insomnia, such as depression and impaired work performance and decision-making, are substantial and lead to overall poor quality of life. Therefore, sleep disorders significantly impair social and occupational function, thereby increasing socioeconomic burden [11,12].

Treatment of sleep disorders such as insomnia largely includes drug therapy and non-medical therapy. Medications such as benzodiazepines, barbiturates, antidepressants, and melatonin agonists are the easiest to use and fastest to act; however, there are concerns about adverse events arising from drug overuse, such as resistance with long-term use, reactions in the event of disruption, and withdrawal symptoms [13]. Non-pharmacological therapy for insomnia consists of cognitive behavioral therapy for insomnia (CBT-i), which is a type of therapy used for behavior and lifestyle changes, and complementary medicines including natural sleep aids and herbal remedies are used. Recently, interest in alternative medicines such as medicinal plants and dietary supplements has been growing, as although medication treatment has a fast and good effect on insomnia, there is a high potential for abuse and dependence [14].

Many medicinal plants, including herbal remedies, are good sources for the treatment of various health problems in humans. Natural plants such as chamomile and valerian are known alternatives to drugs that induce sleep in patients with sleep disorders. Generally, the mechanisms of medicinal plants in treating insomnia are mostly related to gamma-aminobutyric acid (GABA) neurotransmission that regulates sleep outcomes. In addition, such products are usually associated with the production of the sleep control hormone melatonin. In recent years, with increasing interest in sleep induction and maintenance, new, effective, and safe sleeping pills have been developed. In particular, there has been growing interest in considering the use of melatonin as a sleeping hormone in patients with sleep disorders as it does not induce hallucinations or result in dependence.

Melatonin (N-acetyl-5-methoxytryptamine) is an endogenous hormone produced by the pineal gland and released into the bloodstream. Melatonin has various beneficial effects on sleep and circadian abnormalities, mood disorders, learning and memory, neuroprotection, drug abuse, and cancer by activating the high-affinity G protein-coupled receptors MT1 and MT2 [15,16,17,18]. Melatonin synthesis is controlled by endogenous oscillators within the suprachiasmatic nucleus (SCN) and is regulated by daily and seasonal changes in the environmental light–dark cycle [18,19]. Melatonin synchronizes with circadian rhythms and improves sleep onset latency, duration, and quality [20]. Additionally, it has been shown to have antioxidative, anti-inflammatory, sleep-regulatory, and neuronal survival effects. Melatonin and melatonin agonists play a crucial role in the treatment of insomnia by activating the MT1 and MT2 melatonin receptors [18]. Exogenous melatonin can be effective in the deficiency of melatonin release by mimicking the natural endogenous melatonin, binding to MT1 and MT2 receptors, and activating the same downstream pathways. Ramelteon, a melatonin receptor agonist, promotes sleep by acting on MT1 by decreasing wake-promoting signals and might influence sleep duration via MT2. Exogenous melatonin supplementation and melatonin agonists had no clear short- or long-term adverse effects. In this regard, exogenous melatonin and melatonin agonists acting on functional melatonin receptors are considered potential targets for the treatment of sleep disorders, learning and memory deficits, neurodegeneration, and drug addiction. However, the development of sleeping supplements with high efficiency, low dependency, and few adverse effects is still required.

*Maydis stigma* (Corn silk) is composed of silk threads from the stigma of corn fruits and consists of various chemicals, including proteins, vitamins, alkaloids, tannins, mineral salts, carbohydrates, steroids, and flavonoids, as well as volatile chemicals [21,22]. Corn silk is a medicinal plant that has been used in traditional Chinese medicine for a long time and is used to treat several diseases. It has been reported to have multiple beneficial effects, including hypotensive, antidiabetic, antifatigue, antidepressant, and hypolipidemic properties [23,24,25,26]. In addition, it has excellent antioxidant and neuroprotective effects [27]. Corn silk and its bioactive components in ethanol and methanol extracts provide many benefits and can potentially be exploited for use in healthcare applications. There are various commercial products made from corn silk that are available on the market for medicinal use. However, the potential for sleep improvement of *Maydis stigma* remains unknown. In the present study, we investigated the effects of *Maydis stigma* on sleep-promoting properties and possible signaling mechanisms targeting MT1 and MT2 melatonin receptors.

## 2. Results

### 2.1. Maydis Stigma Extracts Increased mRNA Expression of Melatonin Receptors 1 and 2

We investigated the effects of *Maydis stigma* extract (MSE) on melatonin receptor 1 (MT1) and melatonin receptor 2 (MT2) expression in rat primary cortical and hypothalamic neurons. Cultured cortical and hypothalamic neurons were treated with 5 and 25 μg/mL MSE on DIV 8. After 24 h, RNA was isolated from cultured cortical neurons using TRIzol reagent. The isolated RNA was quantified, and qRT-PCR was performed to synthesize cDNA (0.5 μg). Melatonin receptor 1/2 mRNA expression was measured by PCR using the melatonin receptor 1/2 primer set. Our data showed a 3.3- to 4.4-fold increase in MT1 expression and a 1.7~2-fold increase in MT2 expression compared to the control hypothalamic neurons, and a 2.2-fold increase in MT1 expression and a 1.9-fold increase in MT2 expression compared to the control in rat primary cultured cortical neurons (Figure 1).

### 2.2. MS Extracts Reduced Sleep Onset Latency and Increased Sleep Duration

We used a pentobarbital-induced sleep model to investigate the effects of MSE on sleep regulation. Mice (4 weeks old) were orally administered MSE (1, 10, and 100 mg/kg) for 7 days, and water was administered as the vehicle. Valerian root extract (10 mg/kg) was used as a positive control. After 7 days, all experimental mice were intraperitoneally injected with 42 mg/kg pentobarbital sodium, and the time to sleep onset and awakening was measured. The onset time of the MSE-treated group was 30 s earlier than that of the vehicle control group (202.1 ± 9.6 s), and the time of sleep maintenance was approximately 1000 s longer than that of the control group (2043.7 ± 147.8 s). These results showed that MSE mice fell asleep faster and maintained sleep longer than vehicle-treated mice in a dose-dependent manner (Figure 2). In addition, valerian root extract-treated mice showed decreased sleep onset time and increased sleep duration. Interestingly, our data demonstrated that MSE treatment had a sleep-inducing and sleep-maintaining effect.

We examined whether MSE administration increased the mRNA and protein expression of MT1/2 in the brains of these animals. MSE administration significantly increased the mRNA expression of MT1 and MT2 (1.7- and 2.4-fold, respectively) in the hypothalamus and MT1 (~1.8-fold) in the frontal cortex of the brain compared to that in the vehicle-injected group (Figure 3A,B). In addition, MSE administration significantly increased the protein expression of MT1 (1.56-fold) and MT2 (1.84-fold) in the frontal cortex of the brain compared to that in the vehicle control (Figure 3C). To evaluate whether MSE administration increased melatonin secretion, melatonin levels were measured in the blood of mice after MSE administration. Compared with the vehicle control, the plasma melatonin level was significantly increased in the MSE administration group (~1.36-fold). The results indicate that MSE can effectively increase melatonin levels.

We also evaluated the effect of MSE on caffeine-induced sleep disturbance using a pentobarbital-induced sleep test in mice. Sleep disturbance was induced by the administration of caffeine (10 mg/kg, i.p.) 30 min before the administration of sodium pentobarbital (42 mg/kg, i.p.). Caffeine-treated mice (237.2 ± 14.8 s) exhibited a significantly increased time to sleep onset compared with the vehicle control (181.9 ± 10.1 s). Moreover, caffeine-treated mice (1523.0 ± 175.9 s) showed significantly lower sleep duration than the vehicle control (2569.0 ± 218.5 s). Mice that were pretreated with MSE (1, 10, and 100 mg/kg), however, significantly recovered from the caffeine-induced increase in sleep latency (195.2 ± 5.6, 174.5 ± 5.9, and 161.3 ± 7.5 s, respectively), and 100 mg/kg MSE-treated mice significantly recovered from the caffeine-induced decrease in sleep maintenance (1983.6 ± 113.4 s) (Figure 4). These results demonstrated that MSE improves sleep latency and sleep duration in a sleep disturbance model as well as a sleep-inducing model.

### 2.3. MS Extracts Increased the Expression of Circadian Clock Genes

Previous studies showed melatonin directly acts on the SCN regulating the circadian rhythms. A functional MT1 receptor is also required for the regulation of the expression of clock genes such as *Bmal1* and *Clock* in the brain [28,29]. The regulation of circadian rhythms is based on the transcriptional feedback loop of clock genes including *Per*, *Cry*, *Clock*, and *Bmal1*. To examine the molecular mechanism of MSE related to circadian clock genes, we determined the mRNA expression of circadian clock-regulated markers including *Per*, *Cry*, *Clock*, and *Bmal1* using qRT-PCR. Our results showed that MSE (100 mg/kg) significantly increased the expression of clock genes including *Per1/2*, *Cry1/2*, and *Bmal1* in the hypothalamus (3.3/3.78-, 2.07/2.37-, and 2.46-fold, respectively) compared to vehicle control (Figure 5). These results demonstrated that MSE regulates the expression of circadian clock genes via activation of melatonin receptors, thereby improving sleep quantity and quality.

### 2.4. Butanol Fraction of MSE Reduced Sleep Onset Latency and Increased Sleep Duration

To investigate the nature of compounds responsible for the effects of MS extracts, three fractions were prepared: the water fraction (WF) solubilizing the polar agents and water-soluble plant constituents (e.g., glycosides, quaternary alkaloids, tannins), ethyl acetate fraction (EAF) extracting compounds of intermediate polarity, and butanol fraction (BuF) bearing non-polar agents such as sterols, alkanes, and some terpenoids [30]. When BuF was administered, the latency time decreased from 172.4 ± 23.47 (vehicle) to 145.45 ± 26.95 s (*p* < 0.05), and the sleep duration increased from 2299.5 ± 188.3 to 3466.76 ± 169.6 s (*p* < 0.001). Our data showed that the butanol fraction (BuF) of MS extracts significantly prolonged sleep duration and decreased sleep latency (Figure 6). The fact that BuF potentiates sleep parameters indicates that non-polar agents may be responsible for the effects of MS on sleep improvement.

### 2.5. β-Sitosterol Had a Sleep-Promoting Effect

Existing evidence supports the possible role of phytosterols and non-polar components. In particular, β-sitosterol, which is a component of corn silk, is widely known as the major active compound and indicator constituent in corn silk extracts [26]. The hypnotic effects of β-sitosterol have been reported previously. β-Sitosterol has sedative and sleep-promoting effects via GABA transmission [31,32].

To investigate the effects of β-sitosterol on sleep regulation, we performed a pentobarbital-induced sleep test. Mice were administered β-sitosterol BS (0 and 100 mg/kg) and saline as a vehicle for 7 days. As expected, sleep onset latency (248.73 ± 16.18 s, 247.3 ± 12.93 s, respectively, *p* < 0.05 vs. vehicle control) was significantly reduced by BS, and sleep duration time (2764.5 ± 64.15 s, 2983.63 ± 200.85 s, respectively, *p* < 0.05 vs. vehicle control) was increased compared to the vehicle control (389.21 ± 42.48, 1836.62 ± 49.61 s, respectively). Diazepam (positive control, 2 mg/kg) decreased the pentobarbital-induced sleeping onset time and increased the sleeping duration time (173.44 ± 15.61 s, 3856.67 ± 387.66 s, respectively, *p* < 0.01 vs. vehicle control) (Figure 7). This result showed that β-sitosterol, the major component of corn silk, significantly reduced sleep latency time and increased sleep duration.

We examined whether β-sitosterol treatment increased the mRNA expression of MT1/2 in the brains of these animals. β-Sitosterol administration significantly increased the mRNA expression of MT1 and MT2 (1.5- and 3.8-fold, respectively) in the cortex of the brain compared to that observed in the vehicle-treated group (Figure 7). Our results suggest that MS affects sleep improvement; in particular, β-sitosterol, as a non-polar agent of the butanol fraction, can be responsible for the effects of MSE.

### 2.6. Effects of MS Extracts and β-Sitosterol on the EEG Waves

To investigate the effects of MS extracts and β-sitosterol on the brain activity of sleep behavior, we recorded an electroencephalogram (EEG). EEG signals in mice were recorded for 40 min after oral administration of MSE and β-sitosterol and for 60 min after pentobarbital sodium injection. Our result showed the difference between vehicle and MSE/β-sitosterol-treated mice in EEG trace during treatment. Alpha waves, which increase in the resting stage before sleep, increased significantly in both groups after administration of pentobarbital sodium, and in particular, they increased more significantly (80%) in the group treated with 20 mg/kg of MSE than in the vehicle control group during treatment for 7 days. In addition, the 10 mg/kg of β-sitosterol-treated group showed significantly increased alpha waves (450%), and the 2 mg/kg of diazepam (DZP)-treated group also showed increased alpha waves compared to the vehicle group (278%) (Figure 8). However, MSE and β-sitosterol did not affect other EEG waves during the experimental period (data not shown). An increase in alpha waves indicated a decrease in sleep latency. These results suggest that the MSE as well as β-sitosterol can decrease the sleep latency and can lead to accelerating the initiation of non-rapid eye movement sleep (NREMS).

### 2.7. MS Extracts Do Not Cause Place Preference

To investigate the effects of MS extracts on addiction, we performed the conditioned place preference (CPP) test. Figure 9 shows the time spent in the initially non-preferred area (drug-paired compartment) of the apparatus during the pre- and postconditioning phases of the CPP test (*p* < 0.0384). These data are represented as CPP score, which is the difference in the time spent in the saline-paired or drug-paired compartment during the pre- and postconditioning. These data showed that mice conditioned using 1 mg/kg of methamphetamine had a significant CPP score, whereas mice treated with MS extracts had results similar to those of the vehicle group in terms of the time spent in the drug-paired compartment.

### 2.8. MS Extracts Inhibited the H_2_O_2_-Induced Neuronal Cell Death in Rat Cortical Neurons

We tested the protective effects of MS extract on H_2_O_2_-induced neuronal cell death in rat primary cortical neurons. After 24 h of treatment with *Maydis stigma* extracts, 50 M H_2_O_2_ was added. After 24 h, cell viability was measured using an MTT solution. Neurons cultured with H_2_O_2_ (50 μM) decreased cell viability by 10.7% compared to vehicle-treated cells (100%). However, pretreatment with MSE (0.2, 2, and 20 μg/mL) increased cell viability in a dose-dependent manner, by 41.6%, 62.4%, and 66.6%, respectively, compared with vehicle-treated control cells. NAC (N-acetyl cysteine, 100 μM), an antioxidant and neuroprotectant, increased cell viability by 34.9% compared to the vehicle control (Figure 10). These results indicate that MSE can protect neuronal cells against oxidative stress-induced neuronal cell death.

### 2.9. Both MS and BS Regulated the Phosphorylation of ERK and αCaMKII in Rat Primary Cultured Cortical Neurons

To investigate the signaling molecules associated with sleep-promoting effects, we tested the association between ERK1/2 and αCaMKII using Western blot analysis. The cells were treated with MSE (0.2, 2, 20, and 200 μg/mL) and BS (0.1, 1, 10, and 100 M) for 30 min. MSE (200 μg/mL) treatment increased ERK1/2 phosphorylation by 143.2% and αCaMKII phosphorylation by 147.1% in rat primary cultured cortical neurons compared to vehicle-treated control cells (100%) (Figure 11). BS (100 μM) also increased ERK1/2 phosphorylation (169.1%) and αCaMKII phosphorylation (411.8%) compared to vehicle-treated control cells (100%). These results showed that both MSE and BS activated the phosphorylation of ERK1/2 and CaMKII, which are associated with downstream signals of melatonin receptors.

## 3. Discussion

Many people are suffering from sleep deprivation, and the prevalence of sleep deprivation has been increasing with rapid social and economic development. Therefore, many efforts have been made to develop nutraceuticals and functional foods as a method for treating sleep disorders, as well as chemical drugs without toxicity and adverse effects such as addiction. They are usually thought to be related to the production of melatonin, a sleep control hormone, and its receptors such as melatonin receptors 1 and 2. In this study, we demonstrated that MS extracts could exert a protective effect against neurotoxicity, increase the expression of melatonin receptors 1 and 2 and clock genes in the brain, decrease sleep latency, and increase sleep duration. We also investigated the compounds responsible for the effects of MS extracts. Our results revealed that the butanol fraction (BuF) with non-polar components, including β-sitosterol, significantly prolonged sleep duration and decreased sleep latency. In addition, our results showed that β-sitosterol also has an effect on sleep improvement, consistent with previous studies that showed that β-sitosterol has sedative and sleep-promoting effects [31,33]. All of these results suggest that MS extracts can be a novel therapeutic candidate for sleep disorders such as insomnia, as well as other neurological disorders related to sleep disorders. To the best of our knowledge, this study is the first to investigate the novel use of MSE on sleep improvement and the melatonin receptors as target molecules.

Most studies on sleep deprivation were performed mainly in male subjects or did not consider gender factors, although sleep deprivation is correlated with gender-specific differences and several clinical studies have shown sex differences in insomnia and other sleep pathologies [34]. Koehl et al. suggested that there was no difference between genders in mice experiments on sleep deprivation [35]. However, female rodents, like humans, show changes in sleep and locomotor activity across their estrous cycle [36]. In addition, sleep studies in rodents showed that sex differences in sleep were eliminated by the removal of sex hormones and gonadectomy. Therefore, in this study, we only used male mice as experimental subjects to avoid the effects of the estrous cycle.

*Maydis stigma* (MS, corn silk) is known to have the potential to be developed into a functional food because of its various functions. Corn silk contains various components, including flavonoids (maysin, methoxymaysin, apimaysin, and luteolin derivatives), saponins, tannins, phenols, alcohols, terpenoids, glycosides, proteins, carbohydrates, vitamins, alkaloids, mineral slats, anthocyanins, protokatekin, quercetin, β-carotene, stigmasterol, and β-sitosterol, as well as volatile chemicals [23,37]. For a long time, corn silk has been traditionally used as a treatment for various diseases such as obesity, cystitis, gout, kidney-related diseases, neurodegenerative diseases, and diabetes [27,28,38,39]. In addition, recent studies showed that corn silk exhibits antioxidative, antidepressant, and anti-inflammatory activities [25,40]. In this study, our results showed that MS extracts decrease sleep latency and increase sleep duration. Furthermore, these results demonstrated new properties of *Maydis stigma*, corn silk, namely its sleep-promoting effects.

Melatonin is synthesized and released by the pineal gland. The physiological actions of melatonin are mediated by two G-protein-coupled membrane receptors, MT1 and MT2 [41,42,43]. Among its various beneficial effects, the best known is that it plays a crucial role in circadian rhythmicity, which has a sleep-promoting property [44,45]. Melatonin enhances GABAergic inhibitory transmission by increasing both the amplitude and frequency of GABAergic mIPSCs, which may be a potential pathway for the neuroprotective and sleep-promoting effects of melatonin [46]. Melatonin receptors comprise heterodimers that modulate signaling pathways; are distributed throughout the brain and many different peripheral organs; and modulate multiple functions associated with sleep, diabetes, depression, autism spectrum disorders, and many neurodegenerative disorders such as Alzheimer’s disease (AD) and Parkinson’s disease (PD) [47,48]. The high-affinity MT1 and MT2 receptors have distinct functional roles in sleep regulation. Activation of the MT1 receptor suppresses the neuronal firing rate in the SCN, while MT2 acts mainly by inducing circadian rhythm phase shifts. In addition, activation of the MT1 melatonin receptor regulates *Per1* gene expression in the anterior pituitary and mediates vasoconstriction in the cerebral and peripheral arteries. Activation of the MT2 melatonin receptor phase shifts circadian rhythms generated within the SCN and inhibits dopamine release in the retina [49]. Recently, a study on the differential function of the MT1 and MT2 receptors using MT1 and MT2 receptor knockout mice suggested that MT1 receptors primarily regulate rapid eye movement (REM) sleep, whereas MT2 receptors increase non-REM (NREM) sleep using MT1 and MT2 receptor knockout mice [42]. The localization of MT1 receptors is distinct from that of MT2 receptors. MT1 receptors are expressed in the locus coeruleus and lateral hypothalamus, whereas MT2 receptors are expressed in the reticular thalamus. Our data showed that MSE increased the expression of MT1 and MT2 melatonin receptors in the cortex and hypothalamus, demonstrating that MSE can improve sleep disturbances by regulating NREM and REM sleep.

Circadian rhythms in physiology and behavior are regulated by a central clock located within the SCN of the hypothalamus. The circadian clock is controlled by light and melatonin, regulates sleep and wake cycles, and has been shown to promote alertness during the day [50]. The biological clock is based on the transcriptional/translational feedback loop of clock genes (*Per*, *Cry*, *Clock*, and *Bmal1*). von Gall et al. reported that *Per1*, *Cry1*, *Clock*, and *Bmal1* were reduced in MT1 KO mice but were expressed in WT and MT2 KO mice. Therefore, melatonin, acting through the MT1 receptor, is an important regulator of rhythmic clock gene expression in the pars tuberalis (PT) of the pituitary [28,29]. Furthermore, melatonin treatment induces the clock gene to reverse erythroblastosis virus (REV-ERBα) expression in the rat hypothalamus [51]. In this study, our results showed that MSE increased the expression of clock genes, including *Per1/2*, *Cry1/2*, and *Bmal1*, with an increase in melatonin receptor expression. von Gall et al.’s report [28] supports our findings that the increase in melatonin receptor expression is associated with the expression of clock genes. Therefore, these results suggest that MSE has a sleep-promoting effect through an increase in the expression of melatonin receptors and clock genes, and these effects may be caused by an increase in melatonin release. All of these results demonstrated that MSE has a novel effect on sleep improvement and that this effect may be associated with the activation of clock genes as well as melatonin receptors, thus showing that MSE can be a potential candidate for treating sleep disorders related to sleep loss and poor sleep quality. However, further studies are needed to elucidate the mechanisms by which clock genes and melatonin receptors are regulated by MSE.

There is growing evidence of the potential role of melatonin receptors in the modulation of sleep and circadian rhythms [18,52], and melatonin agonists such as Circadin, agomelatine, and tasimelteon are currently used clinically in the treatment of insomnia. Circadin is used to treat primary insomnia in patients, especially those with poor sleep quality [53]. Agomelatine and tasimelteon, which are melatonin receptor 1/2 agonists, improved sleep quality and reduced waking after sleep onset [54,55]. Additionally, there are many studies on the distinct roles of MT1 and MT2 receptors in sleep–wake regulation [56,57,58]. Therefore, the discovery of selective ligands targeting MT1 and MT2 receptors may facilitate the development of novel and more efficacious therapeutic agents for sleep disorders and related neurodegenerative diseases [18,59]. In this study, we investigated the potential role of corn silk, MS, by targeting MT1 and MT2 melatonin receptors for the treatment of sleep disorders. Our data showed MSE increased melatonin receptor 1 and 2 expression, which is a possible role in affecting the sleep–wake cycle regulation. Therefore, our data suggest that *Maydis stigma* could be a candidate material for developing novel sleep therapeutics based on melatonin receptors.

We tested three fractions of MSE to determine the nature of the compounds responsible for the effects of MSE. A water fraction (WF), a butanol fraction (BuF), and an ethyl acetate fraction (EAF) of MSE were prepared. The water fraction (WF) solubilizes polar agents and water-soluble plant constituents (for example, glycosides, quaternary alkaloids, and tannins). The ethyl acetate fraction (EAF) is a compound of intermediate polarity (e.g., some flavonoids), and the butanol fraction (BuF) contains non-polar agents, such as sterols, alkanes, and some terpenoids [60]. Among the WF, EAF, and BuF fractions, BuF not only prolonged sleep duration but also induced a significant decrease in sleep latency. Our data showed that the BuF of MS extract significantly enhanced sleep duration and decreased sleep latency (Figure 5). However, the EAF did not affect sleep latency or sleep duration. These results show that the BuF contains a higher concentration of constituents responsible for the hypnotic effect of MS. The fact that the BuF potentiates sleep parameters indicates that non-polar agents such as β-sitosterol may be responsible for the effects of MS on sleep improvement.

Many phytochemicals have been reported to exert sedative–hypnotic effects. They contain terpenoids (e.g., linalool, eugenol), flavonoids (e.g., quercetin and luteolin), alkaloids (e.g., rosmarinic acid), sterols (e.g., β-sitosterol), and saponins. β-Sitosterol is a phytosterol; phytosterols mainly include sitosterol, stigmasterol, and campesterol, which are essential steroid molecules that stabilize cell membranes in plants [61,62]. β-Sitosterol (BS) is a phytosterol that has structural and biological functions similar to those of cholesterol and is a member of a major group of bioactive components with well-confirmed bioactivity. β-Sitosterol is considered a safe and effective nutritional supplement with various potential benefits. BS has been shown to have antioxidant, antimicrobial, anti-inflammatory, antinociceptive, and antidiabetic activities. Therefore, there are some nutraceutical materials that contain BS on the market. Neuroactive steroids such as BS modulate GABA_A_ receptors, which have potential anticonvulsant, anxiolytic, neuroprotective, and sedative–hypnotic effects [63,64]. It has been reported that BS improves sleep via GABAergic systems [31], and BS and its derivatives showed antidepressive effects by modifying 5-HT, DA, and GABAergic systems in mice [33]. Our results demonstrate that BS, a major component of MSE, has a sleep-promoting effect, which increases the expression of melatonin receptors 1 and 2. Therefore, our results suggest a novel mechanism of BS on sleep improvement targeting melatonin receptors.

In this study, we first found a remarkable hypnotic effect of MS and β-sitosterol through a pentobarbital-induced sleep test and caffeine-induced sleep disturbance test, along with EEG analysis. Our results showed that there was a difference in EEG traces between vehicle- and MSE/β-sitosterol-treated mice, and the administration of MSE and β-sitosterol increased alpha wave activity. Among EEG sleep waves, alpha waves increase during the resting stage before sleep. Both MS extracts and β-sitosterol administration significantly increased alpha waves after the administration of pentobarbital sodium. However, during the experimental period, the other waves of EEG were not changed by MSE or β-sitosterol treatment. An increase in alpha waves indicates a decrease in sleep latency, and alpha waves are closely related to memory, creativity, and academic performance [65,66]. The EEG activity of alpha and theta waves is associated with cognitive and memory performance [67]. In addition, alpha waves increase sleep induction, and it has been shown that alpha wave activity is related to non-rapid eye movement (NREM), a deeper sleep state, at the beginning of sleep [68]. Therefore, our results suggest that MSE as well as β-sitosterol can induce sleep and accelerate the initiation of NREM sleep. However, further investigation is needed in future studies to clarify the connection between MSE and sleep.

Pharmacological treatments have been used for various types of sleep disorders such as insomnia. Current medications including benzodiazepines, barbiturates, and melatonin agonists should be prescribed by a doctor due to extensive adverse effects, such as dependence and abuse [69]. BZDs and barbiturates are classified as Schedule IV controlled substances by the US Drug Enforcement Administration (DEA). BZDs are frequently prescribed to treat insomnia because they reduce sleep latency and increase total sleep duration by enhancing the inhibitory action of GABA. However, BZDs have severe side effects such as rebound insomnia and anterograde amnesia. In addition, the accumulation of active metabolites can cause confusion, cognitive dysfunction, and depressive features [70] and can produce residual sleep medication effects (e.g., drowsiness, difficulty concentrating, and impaired memory) and interfere with quality of life [71]. Ramelteon, as a melatonin agonist, can be used to treat insomnia and is the only non-scheduled prescription drug in the United States. Second-line medications include antidepressants and antihistamines that have moderate levels of tolerability. Some antidepressants used to treat insomnia are associated with increased suicidal ideation, increased mania/hypomania, and exacerbation of restless leg syndrome. Further, anticonvulsants can produce daytime sedation, dizziness, and cognitive impairment. Elderly individuals are particularly vulnerable to residual sleep medication effects. Along with the increase in the elderly population, the number of people suffering from sleep disorders is also increasing. Therefore, there has been an increased interest in finding a new approach to sleep problems with fewer adverse effects. Based on this reason, there is increasing interest in nutraceuticals, which are safe even if taken for a long time, for the treatment of sleep disorders [14].

“Nutraceutical” is a combination of two words, “nutrition” and “pharmaceutical”, and is defined as part of food that provides medical and health benefits. Nutraceuticals include herbal products, isolated nutrients, and dietary supplements, which are used as a potential treatment for various diseases due to their non-toxic properties. Recently, herbal medicine has been used to treat sleep disruption through its anxiolytic and sedative properties worldwide. Herbal medicines consist of mixtures of compounds, and their mechanisms of action often remain unclear. However, several studies have shown that the anxiolytic and sedative effects of herbal plants are exhibited through GABAergic neurotransmission, similar to pharmacological treatments such as benzodiazepine receptor agonists [72]. In this perspective, we tested whether MS extracts result in dependency using the CPP test. These results showed *Maydis stigma* did not result in dependency, and thus it can be a potential candidate as a nutraceutical for treating sleep disorders such as insomnia. While we recognize the limitation of how an animal model can fully recapitulate human physiology, our study provides insights on how *Maydis stigma* affects sleep and opens a new avenue for further investigation.

In the present study, our data showed that MS extracts provided protection against oxidative stress-induced neuronal cell death. This neuroprotective effect may be associated with an increase in antioxidant enzymes such as CAT and SOD. Recently, maysin, a flavone glucoside isolated from corn silk, was shown to be neuroprotective against oxidative stress-induced apoptotic cell death [38]. Previously, corn silk maysin prevented apoptotic cell death by inhibiting DNA damage and PARP cleavage and inhibited oxidative stress by increasing the levels of antioxidant enzymes, including CAT, GPx, and SOD [38,73].

We investigated the signaling molecules of MSE that are associated with sleep-promoting effects. Our results showed an increase in MT1/2 expression following treatment with MSE (Figure 1). MT1 and MT2 receptors have been shown to activate several signaling molecules, including ERK1/2 and PKC. Melatonin stimulation activated MT1 and MT2 receptors coupling to the Gq/PLC/Ca^2+^ pathway and Gi/cAMP pathway [43,74]. Other major pathways activated by melatonin are the ERK1/2 and MEK1/2 pathways. These signaling pathways likely mediate the effects of MSE on sleep improvement. Although there are various intracellular molecules for the underlying molecular mechanisms through melatonin receptor activation, we focused, as a first step, on the major protein kinase in the brain, calcium/calmodulin-dependent protein kinase type II (CaMKII), and ERK1/2, which is known to be able to interact with some of the Ca^2+^-dependent pathway [75,76,77]. Several studies have reported the modulation of sleep behavior by ERK activation. The upregulation of ERK1/2 activity is critical in the regulation of REM sleep [78,79], and cortical ERK activation is involved in sleep consolidation in cats [80]. In addition, it has been reported that impairment of CaMKII genes, such as *Camk2a* and *Camk2b*, decreases sleep duration [76]. In this study, our data showed that MSE increased ERK1/2 and CaMKII phosphorylation, consistent with previous reports. Therefore, our results demonstrated that CaMKII and ERK1/2 are candidate molecules underlying the sleep-promoting effects of MSE.

## 4. Materials and Methods

### 4.1. Maydis Stigma Extract Preparation

Corn silk (*Maydis stigma*) extract was supplied by the Korean Plant Extract Bank of the Korea Research Institute of Bioscience and Biotechnology. For this extraction, 100 g of corn silk was mixed with 1 L of water and then heated at 100 °C for 2 h and 30 min. The extracted material was filtered using a cotton pad filter paper and dried using a vacuum freeze dryer (Biotron, Clean-vac 12) at −70 °C for 24 h. The dried extract was stored at −4 °C until use. To test melatonin receptor expression, *Maydis stigma* extracts were dissolved in 100% dimethyl sulfoxide (DMSO).

After extraction, *Maydis stigma* extracts were concentrated at 80–90% and freeze-dried. The extraction yield was found to be 13%. Thereafter, the fractionation of lyophilized extracts was performed in the order of ethyl acetate, n-butanol, and DW (EnsolBio Sciences Inc., Daejeon, Korea) [30,81].

### 4.2. Animals

All procedures involving animals in this study were conducted in accordance with animal care and use guidelines approved by the Animal Care and Use Committee of Konkuk University (KU19074). Every effort was made to minimize the number of animals used and any pain or stress they might experience. Male ICR mice (20–30 g; four weeks old) were purchased from Orient Bio (Gyeonggi, Korea) and used for the experiment. The animals were housed in a soundproof room maintained at an ambient temperature of 23 ± 0.5 °C with a relative humidity of 55 ± 2% on an automatically controlled 12 h light/12 h dark cycle (lights on at 7:00). All the animals had free access to food and water.

### 4.3. Cell Culture and Drug Treatment

For primary cortical neurons and hypothalamic neuron cultures, pregnant Sprague–Dawley rats were purchased from Orient Bio (Gyeonggi, Korea), and primary cortical neurons were isolated from embryonic day-14 (E14) hypothalamus and day-17 (E17) cortices of SD rat embryos, as described previously [82]. Briefly, the cortex was mechanically dissociated and gently triturated in the culture medium (neurobasal media). The cells were seeded at 2 × 10^7^ cells/mL onto 50 µg/mL poly-d-lysine (PDL)-coated plates in neurobasal medium supplemented with 1% B-27 supplement and 2 mM L-glutamine (Gibco, Thermo Fisher Scientific, Waltham, MA, USA). The cultures were maintained at 37 °C in a humidified 5% CO_2_ incubator, and the culture medium was changed every 2 days. Cultured cells were used after 8 days of in vitro differentiation.

### 4.4. Pentobarbital-Induced Sleep Test

The experimental procedure for sleep evaluation was based on the prolongation of pentobarbital-induced sleeping time [82]. The animals were randomly divided into five groups consisting of 8–12 mice each. All samples were dissolved in water prior to use. All experiments were performed between 13:00 and 17:00 h, and the mice were fasted for 12 h prior to the experiments. In the first experiment, to determine if MSE had a hypnotic effect, the following solutions were administered (p.o.) to mice using an oral Zonde needle 30 min prior to the intraperitoneal (i.p.) injection of pentobarbital: water as vehicle, valerian root extract (10 mg/kg) as a positive control, and MSE (110,100 mg/kg) for 7 days. In the second experiment, four groups of mice were administered (p.o.) with the following agents to determine the most effective fraction: water fraction (WF), butanol fraction (BuF), and ethyl acetate fraction (EAF) (10 mg/kg).

Sleep disturbance was induced by intraperitoneal injection of caffeine (10 mg/kg) 30 min before administering sodium pentobarbital [83]. Following pentobarbital injection (42 mg/kg, i.p.), mice were placed in individual cages and observed for measurements of sleep latency and sleep duration. The observers were blinded to the individual treatments. Mice were considered asleep if they stayed immobile and lost their righting reflex when positioned on their backs. Sleep latency was recorded from the time of pentobarbital injection to the time of sleep onset, and sleep duration was defined as the difference in time between the loss and recovery of righting reflex.

### 4.5. Electroencephalographic (EEG) Recording

EEG was performed to assess brain waves in mice according to previous studies [84]. ICR mice were anesthetized with a combination of Zoletil (50 mg/mL) and Rompun (xylazine 23.32 mg/mL). Materials used for EGG experiments were composed of a tethered, three-channel system head-mount (8200-K3-iS/iSE) securely implanted with two stainless steel screws positioned anteriorly (A/P: −1.0 mm, M/L −1.5 mm) and posteriorly (A/P: −1.0 mm, M/L ± 1.5 mm) and fixed with dental cement. A preamplifier unit was attached to the head mount to provide first-stage amplification (100×) and initial high-pass filtering (0.5 Hz). Signals were transmitted to a data-acquisition system via a tether and commutator. The amplifier/conditioning unit provided an additional 50× signal amplification, high-pass filtering, and an 8th-order elliptic low-pass filter (50 Hz). Signals were adjusted to 400 Hz using a 14-bit A/D converter routed to a computer-based software package via USB (Pinnacle Technology, Inc., Lawrence, Kansas). All efforts were made to minimize animal suffering. Mice were allowed a 7-day recovery period before recording, and food and water were available ad libitum. The mice were habituated to the EEG apparatus for 2 h on two consecutive days (unrecorded). The following day, baseline recording was performed for 2 h. EEG recordings were performed after seven days of treatment. During the recording, habituation was performed for 20 min, followed by treatment with vehicle, MS extracts (20 mg/kg), β-sitosterol (10 mg/kg), and diazepam (2 mg/kg) orally (N = 7–8). After 40 min of recording, an intraperitoneal injection of pentobarbital (42 mg/kg) was administered, followed by a 60 min postinjection recording.

### 4.6. Measurement of Melatonin Levels

After the sleeping test, blood samples were collected from the heart under deep anesthesia. After centrifugation at 1000× *g* at 4 °C for 10 min, separated plasma was withdrawn and stored at −80 °C. Melatonin levels were measured using an enzyme-linked immunosorbent assay (ELISA) kit (Abcam, Cambridge, UK) according to the manufacturer’s protocol.

### 4.7. Conditioned Place Preference (CPP) Test

The CPP test was performed as previously described [85] using the CPP apparatus, a two-compartment polyvinylchloride box measuring 47 × 47 × 47 cm. Each compartment had unique visual and tactile cues. One section had black walls with a smooth black floor, whereas the other section had white painted dotted walls with a rough black floor. A guillotine door separated the compartments during the conditioning phase. A software package (Ethovision, Noldus IT BV, Wageningen, the Netherlands) was used to record animal behavior. The CPP test consisted of three phases: (1) habituation for the first two days and preconditioning on the third day, (2) conditioning, and (3) postconditioning. For habituation, the mice were allowed to access both compartments for 15 min once a day during the first two days. On the third day, the time spent (seconds) in each compartment was measured to determine the preferred and non-preferred compartments of each mouse (preconditioning). The guillotine door was then closed to prepare for the conditioning phase. During this period, experimental mice were placed in the preferred and non-preferred compartments. Mice were injected intraperitoneally with methamphetamine (METH 2.5 mg/kg) or MSE (100 mg/kg) and restricted to their non-preferred compartment. They received saline and were placed in their preferred compartment every alternate day. These treatments were repeated for three cycles (6 days). In contrast, the control group was administered saline daily. Then, for the postconditioning phase, the guillotine door was opened, and drug-free mice were allowed access to both compartments.

### 4.8. Quantitative Reverse Transcription Polymerase Chain Reaction (qRT-PCR)

The expression of melatonin receptors 1 and 2 in treated cortical and hypothalamic neurons and brain tissue was determined by qRT-PCR. RNA was extracted using TRIzol reagent (Invitrogen, Carlsbad, CA, USA), and the concentration was measured using a spectrophotometer (Nanodrop Technologies, Wilmington, NC, USA). cDNA synthesis was performed using 0.5 µg of total RNA and an RT reaction mixture containing RevertAid Reverse transcriptase, reaction buffer (Thermo Fisher Scientific, Waltham, MA, USA), and dNTP (Promega, Madison, WI, USA). Gene-specific primer pairs used in this analysis are listed in Table 1.

qRT-PCR was carried out using a 20 µL reaction system containing 5 µL cDNA, 1.2 µL of each primer set, 3.8 µL distilled water (DW), and 10 µL BrightGreen 2x qPCR Master Mix (Applied Biological Materials Inc., Richmond, BC, Canada) on a QuantStudio3 Real-Time PCR System (A28132, Applied Biosystems, Waltham, MA, USA) using the following parameters: (94 °C, 30 s; 60 °C, 1 min; 72 °C, 30 s) × 30 cycles and then 72 °C for 10 min for *MT1* and *MT2*; (94 °C, 30 s; 60 °C, 1 min; 72 °C, 30 s) × 23 cycles and then 72 °C for 10 min for *Gapdh*. All assays, including those for the controls, were performed in triplicate. The expression levels of GAPDH and 18S rRNA were used as internal controls, and the relative expression of each transcript was calculated using the 2^−ΔΔCT^ formula for the fold change, as described in the ABI user guide.

### 4.9. Western Blot Analysis

After treatment with MS extracts (0.2, 2, 20, and 200 µg/mL), cells were harvested in radioimmunoprecipitation assay buffer consisting of 2 mM EDTA, 0.1% (*w*/*v*) SDS, 50 mM Tris-HCl, 150 mM sodium chloride, 1% Triton X-100, and 1% (*w*/*v*) sodium deoxycholate. Total protein was quantified using a BCA assay kit (Thermo Fisher Scientific, Waltham, MA, USA) and boiled for 5 min at 100 °C. Protein (10 μg) from each sample was loaded onto a 10% SDS-polyacrylamide gel and subjected to electrophoresis (SDS-PAGE) at 100 V for 120 min. The separated proteins were then transferred to nitrocellulose membranes for 90 min, and the blots were blocked with 5% skim milk in TBST for 1 h at 25 °C. Blots were then washed with Tris-buffered saline and 0.1% Tween 20 (TBS-T) and incubated with the appropriate primary antibodies, β-actin (1:10,000), MT1/MT2 (1:1000), and p-pERK/ERK/p-αCaMKII/CaMKII (1:1000) overnight at 4 °C. The blots were then washed three times and incubated with horseradish peroxidase (HRP)-conjugated secondary antibodies (Life Technologies, Carlsbad, CA, USA) at room temperature for 60 min. The bands were detected using an enhanced chemiluminescence detection system (iNtRON Biotech, Seoul, Korea) and visualized using a LAS-3000 image detection system (Fuji, Japan). The bands were quantitated using the ImageJ system, and β-actin was used as the loading control.

### 4.10. Measurement of Cell Viability

Rat primary cortical neurons were cultured in 24-well plates, with each treatment group represented in four wells. After treatment with different concentrations of MS extracts (0.2, 2, and 20 µg/mL) and NAC (N-acetyl-L-cysteine; 500 μM) for 24 h, the cells were incubated with 50 μM H_2_O_2_ for an additional 24 h and then evaluated for cell viability using the 3-[4,5-dimethylthiazol-2-yl]-2,5-diphenyl-tetrazolium bromide (MTT) assay. MTT is a water-soluble tetrazolium salt reduced by metabolically viable cells to a colored, water-insoluble formazan salt. MTT (5 mg/mL) was added to the cell-culture medium and then incubated at 37 °C for 2 h in a 5% CO_2_ atmosphere. The MTT-containing medium was then replaced with DMSO, and the absorbance was read at 570 nm using a microplate reader (Spectramax 190, Molecular Devices, Palo Alto, Santa Clara, CA, USA). The percentage of surviving cells was calculated compared to that of the control (untreated cells).

### 4.11. Statistical Analysis

All experimental results are expressed as the mean ± SEM. Statistical comparisons were performed using one-way ANOVA and *t*-test using GraphPad Prism Version 7.01 (La Jolla, CA, USA), and a value of *p* < 0.05 was considered significant.

## 5. Conclusions

Taken together, our results suggest, for the first time, that *Maydis stigma* may have beneficial effects in promoting sleep and increasing sleep quality without noticeable adverse effects such as addiction and toxicity. In addition, our data demonstrated that the sleep-promoting properties of *Maydis stigma* can be associated with melatonin receptor 1/2 and circadian clock genes. As a result, the new prospects of using *Maydis stigma* in promoting sleep and reducing sleep disorders targeting melatonin receptor 1 and 2 may be considered in the development of novel therapeutic agents. Collectively, our study provided some evidence that *Maydis stigma* can be a safe and potential nutraceutical for various psychiatric disorders associated with sleep disturbance, and further investigation would be necessary to validate its clinical efficacy.

## Figures and Tables

**Figure 1 ijms-23-14612-f001:**
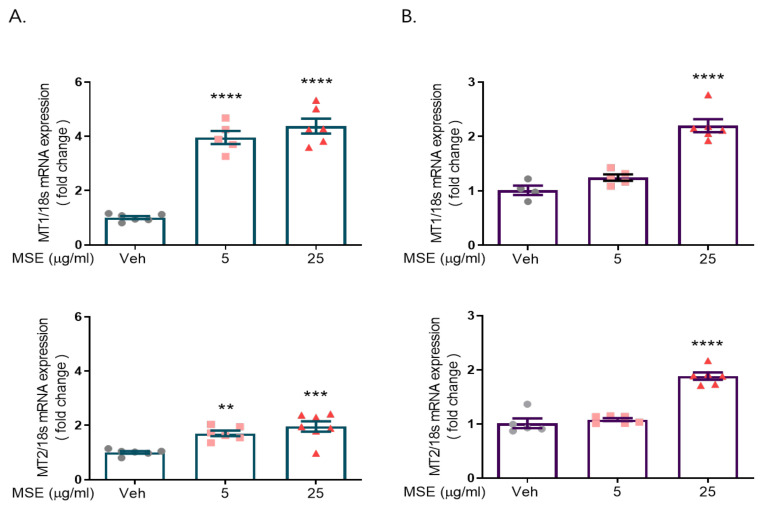
*Maydis stigma* extracts increase mRNA expression of melatonin receptors 1 and 2. Cultured neurons were treated with MSE (5, 25 μg/mL) and positive control (valerian root, VR, 10 mg/kg) on DIV 8. After 24 h, qRT-PCR was performed for measuring the mRNA expression of melatonin receptors 1 and 2 in rat primary hypothalamic neurons (**A**) and rat primary cortical neurons (**B**). ** *p* < 0.01, *** *p* < 0.001, **** *p* < 0.0001; Veh vs. MSE (N = 6).

**Figure 2 ijms-23-14612-f002:**
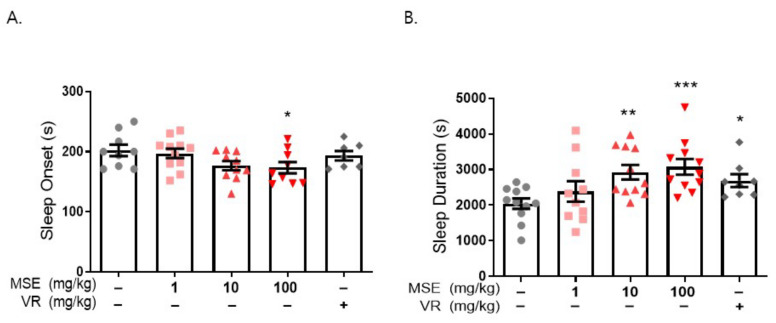
*Maydis stigma* extracts decrease sleep latency and increase sleep duration. Mice were administered MSE (1, 10, 100 mg/kg) and positive control (valerian root extract; VR, 10 mg/kg) for 7 days orally. Sleeping test was performed on day 8. Sleep latency and sleep duration was measured as described in Section 4. (**A**) Sleep latency (s), (**B**) sleep duration (s). * *p* < 0.05, ** *p* < 0.01, *** *p* < 0.001; Veh vs. MSE and VR (N = 8–12).

**Figure 3 ijms-23-14612-f003:**
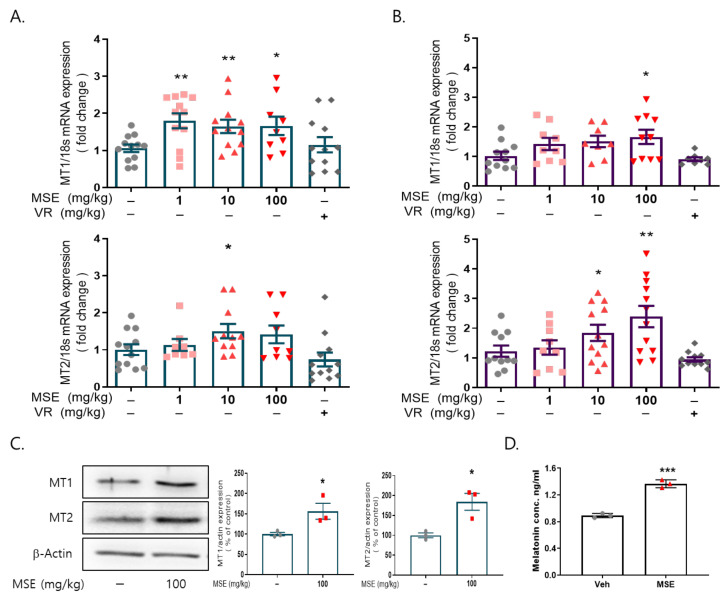
Effects of *Maydis stigma* extracts on the expression of melatonin receptors 1 and 2 in the brain. Mice were administered MSE (1, 10, 100 mg/kg) and positive control (valerian root extract; VR, 10 mg/kg) for 7 days. After sleeping test, blood from the mice was collected, and the brain was dissected. (**A**) mRNA expression of MT1 and MT2 in the hypothalamus, (**B**) mRNA expression of MT1 and MT2 in the frontal cortex. * *p* < 0.05, ** *p* < 0.01; Veh vs. MSE (N = 8–12). (**C**) Protein expression of MT1 and MT2 in the frontal cortex. (**D**) Melatonin levels from plasma. * *p* < 0.05, ** *p* < 0.01, *** *p* < 0.001; Veh vs. MSE (N = 3).

**Figure 4 ijms-23-14612-f004:**
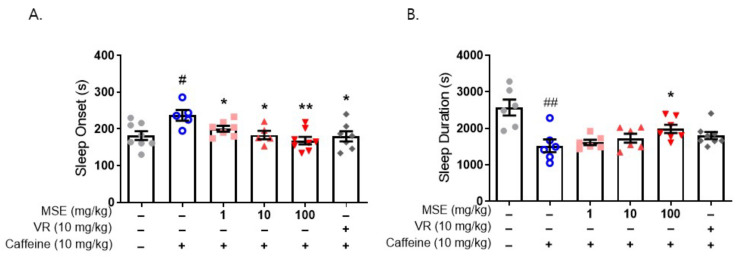
Effects of *Maydis stigma* extracts on caffeine-induced sleep disturbance. Mice were administered MSE (1, 10, 100 mg/kg) and positive control (valerian root extract; VR, 10 mg/kg) for 7 days orally. Sleeping test was performed on day 8. Sleep deprivation was with caffeine (10 mg/kg, i.p.) injection, and then pentobarbital sodium (42 mg/kg, i.p.) injected for sleep induction. Sleep latency and sleep duration were measured as described in Section 4. (**A**) Sleep latency (s), (**B**) sleep duration (s). ^#^
*p* < 0.05, ^##^
*p* < 0.01; Veh vs. Caffeine. * *p* < 0.05, ** *p* < 0.01; Caffeine vs. MSE and VR (N = 5–8).

**Figure 5 ijms-23-14612-f005:**
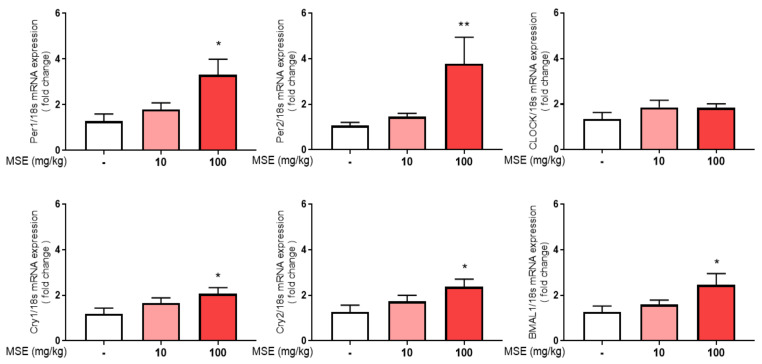
Effects of *Maydis stigma* extracts on the expression of clock genes in the brain. Mice were administered MSE (10, 100 mg/kg) for 7 days. After sleeping test, the mice were sacrificed, and the brain was dissected. qRT-PCR was performed to measure the expression of clock genes including *Per1* and *2*, *Cry1* and *2*, *Clock*, and *Bmal1* in the hypothalamus. * *p* < 0.05, ** *p* < 0.01; Veh vs. MSE (N = 3).

**Figure 6 ijms-23-14612-f006:**
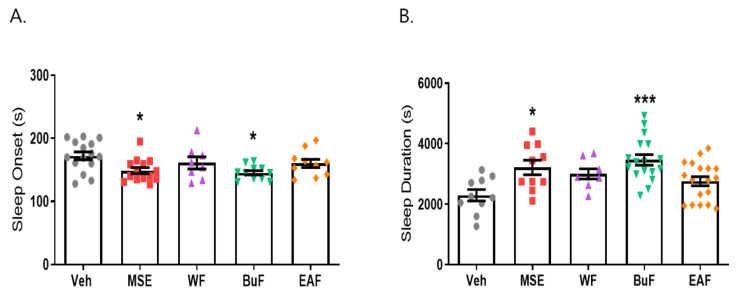
Effects of *Maydis stigma* extract fractions in pentobarbital-induced sleep animal model. Mice were administered the three fractions (WF, BuF, and EAF; 10 mg/kg) of *Maydis stigma* extracts (MSE) and MSE (100 mg/kg) for 7 days orally. Sleeping test was performed on day 8. Sleep latency and sleep duration were measured as described in Section 4. (**A**) Sleep latency (s), (**B**) sleep duration (s). * *p* < 0.05, *** *p* < 0.001; Veh vs. MSE, WF, BuF, and EAF (N = 8–12).

**Figure 7 ijms-23-14612-f007:**
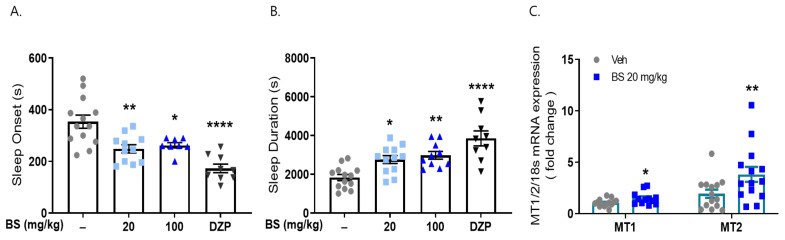
Effects of β-sitosterol on pentobarbital-induced sleep animal model. Mice were administered β-sitosterol (BS, 20, 100 mg/kg) and positive control (diazepam, DZP, 2 mg/kg) for 7 days orally. Sleeping test was performed on day 8. (**A**) Sleep latency (s), (**B**) sleep duration (s). After sleeping test, frontal cortex was dissected, and qRT-PCR was performed for MT1/2 mRNA expression. (**C**) MT1/2 mRNA expression. * *p* < 0.05, ** *p* < 0.01, **** *p* < 0.0001; Veh vs. BS and DZP (N = 9–15).

**Figure 8 ijms-23-14612-f008:**
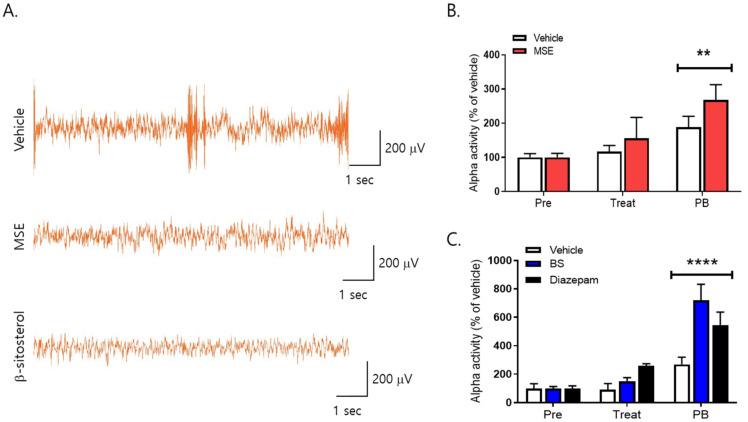
Effects of *Maydis stigma* on alpha waves (8–12 Hz) in mice. Mice were administered *Maydis stigma* extracts (MSE; 20 mg/kg), β-sitosterol (BS; 10 mg/kg), and positive control (diazepam, DZP, 2 mg/kg) for 7 days orally. Each bar represents the mean ± S.E.M. of the total power of alpha waves for 5 min during the habituation period and before and after treatment with pentobarbital sodium (i.p.). (**A**) Representative images of EEG trace during treatment, (**B**) alpha activity after MSE administration, (**C**) alpha activity after β-sitosterol administration. ** *p* < 0.01, **** *p* < 0.0001 vs. Veh (N = 7–8). Pre: habituation; Treat: drug treatment; PB: pentobarbital sodium injection.

**Figure 9 ijms-23-14612-f009:**
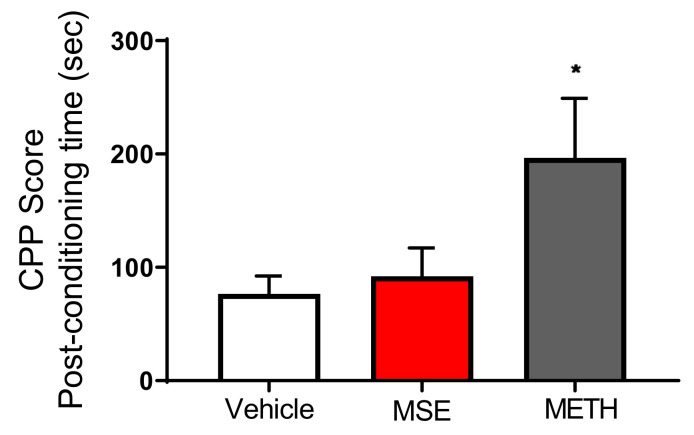
Effects of *Maydis stigma* extracts on place preference in mice. Each bar represents the mean ± S.E.M. of the difference in the time spent in the MSE-, METH-, or Veh-paired compartment during the post- and preconditioning phases of the CPP (N = 10). * *p* < 0.05 vs. Veh.

**Figure 10 ijms-23-14612-f010:**
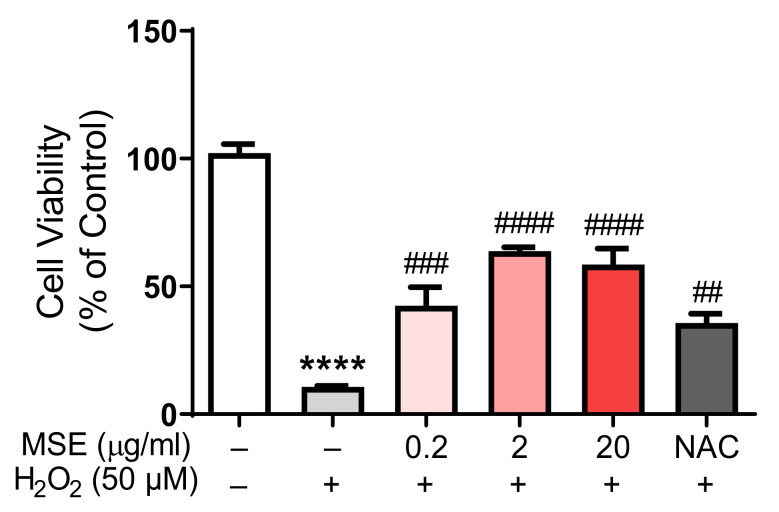
Protective effects of *Maydis stigma* extracts on cell viability in rat primary cortical neurons. MSE (0.2, 2, 20 μg/mL)- and NAC (500 μM)-treated in rat primary cortical neurons on DIV 8. After 24 h, 50 μM of H_2_O_2_ was added to the cultured neurons. MSE treatment prevented H_2_O_2_-induced neuronal cell death. **** *p* < 0.0001; Veh vs. H_2_O_2_. ^##^
*p* < 0.01_,_
^###^
*p* < 0.001, ^####^
*p* < 0.0001; H_2_O_2_ vs. MSE and NAC (N = 6).

**Figure 11 ijms-23-14612-f011:**
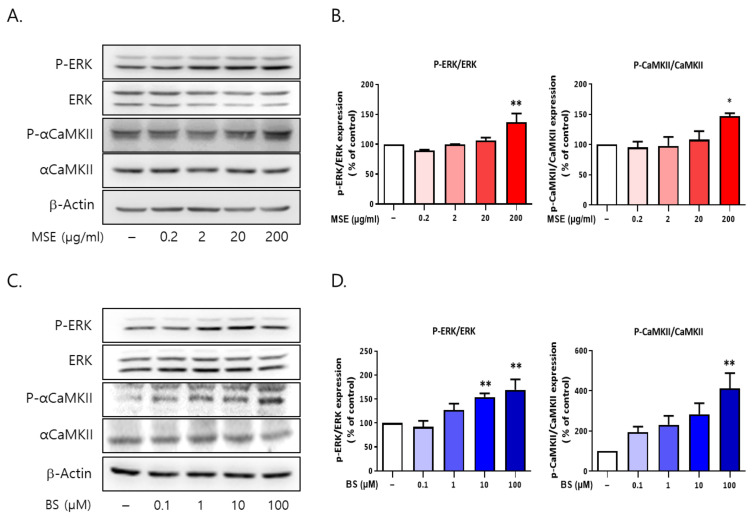
*Maydis stigma* activates the phosphorylation of ERK and αCaMKII in vitro. Cultured cortical neurons were treated with MS extracts (MSE; 0.2, 2, 20, 200 μg/mL) and β-sitosterol (BS; 0.1, 1, 10,100 μM). After 30 min, cells were harvested for measuring the protein expression of p-ERK and p-αCaMKII using Western blot. (**A**,**B**) p-ERK and p-αCaMKII phosphorylation by MSE treatment, (**C**,**D**) p-ERK and p-αCaMKII phosphorylation by BS treatment. * *p* < 0.05, ** *p* < 0.01; Veh vs. MSE and BS (N = 3).

**Table 1 ijms-23-14612-t001:** Gene-specific primers used in this study.

	Sense	Antisense
*MT1*	TTGTGGCGAGTTTAGCTGTG	GACACTCAGGCCCATTAGGA
*MT2*	ATTCCTGCACCTTCATCCAG	CCTGGAGCACCAGTATCCAT
*Per1*	ATGACATACCAGGTGCCGTC	GTCCTCTGAGAACCGTGGC
*Per2*	TCCAGGCACCCGGAATAGTA	CACAACAGGCATGGTGAAGC
*Cry1*	ATGTCCCGAGTTGTAGCAGC	TGAGAGCAATTTCCACCGCT
*Cry2*	TCGAGATACCGGGGACTCTG	GAAGCTGGGCCACTGGATAG
*Bmal1*	CCGTGGACCAAGGAAGTTGA	CTGTTAGCTGCGGGAAGGTT
*Clock*	AGCGATGTCTCAAGCTGCAA	CCTCTATCATCCGTGTCCGC
*18S rRNA*	CATTAAATCAGTTATGGTTCCTTTGG	TCGGCATGTATTAGCTCTAGAATTACC
*Gapdh*	GTG AAG GTC GGT GTG AAC GGA TTT	CAC AGT CTT CTG AGT GGC AGT GAT

## Data Availability

Not applicable.

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
