# Peer review of "Novel Therapeutics for Treating Sleep Disorders: New Perspectives on Maydis stigma"

_ijms, 2022, doi:10.3390/ijms232314612_

Round 1

Reviewer 1 Report

The definition of sleep in the introduction could use some rephrasing – the sensory and muscle activity is present – it can degrease or change otherwise, but it is not inhibited and suppressed.

The literature needs a great update, in the first paragraph, there is a vast number of information on sleep with one completely not adequate citation - Aaby, K., et al., Phenolic compounds in strawberry (Fragaria x ananassa Duch.) fruits: Composition in 27 cultivars and changes during 593 ripening. Food Chem, 2012. 132(1): p. 86-97. 

Further in the introduction still in the first paragraph following the statement “and also, sleep disorder is a common symptom in aging and neuro- 52 degenerative disorder such as AD and dementia, as well as developing disorders such as 53 autism. Sleep disorders produce significant impairments in social and occupational func- 54 tions, which increase the social burden.” There are no references – that need to be corrected.

Describing the therapy for sleep disorders should be much more focused on disorders – the second most common sleep disorder is obstructive sleep apnea – completely different treatment options are used for this disorder. Further, putting benzodiazepines and phenobarbitals on the same scale as an antidepressant is misleading – the mechanisms of the medications are different, and the short-long term effect and need to increase the dose are not similar at all. This narrative seems to be designed to enhance the value of the studied therapeutic – which is again misleading for the reader.

There is no discussion of possible interaction of the used chemical – to induce sleep and alertness as well as the therapeutic – this should be widely discussed as a limitation or described to show a lack of such interactions.

Most of the discussion is just a description that should be in some part provided in the introduction. The discussion of the results compared to available data is very limited compared to the whole section.

Author Response

We appreciate the important points raised by this comment. We believe this comment helped a lot to improve the quality of the manuscript. We added this point throughout the manuscript including introduction, result and discussion.

Point 1. The definition of sleep in the introduction could use some rephrasing – the sensory and muscle activity is present – it can degrease or change otherwise, but it is not inhibited and suppressed.

Response 1: As per your comments, we corrected this sentence as follows.

“Sleep is essential for emotional, physical and cognitive functions. Sleep has restorative effects after activity, allowing optimal functioning [1, 2].” 

Point 2. The literature needs a great update, in the first paragraph, there is a vast number of information on sleep with one completely not adequate citation - Aaby, K., et al., Phenolic compounds in strawberry (Fragaria x ananassa Duch.) fruits: Composition in 27 cultivars and changes during 593 ripening. Food Chem, 2012. 132(1): p. 86-97. 

Response 2: As per your comments, we removed this citation and updated the references in this manuscript. Thank you for your careful comments.

Point 3. Further in the introduction still in the first paragraph following the statement “and also, sleep disorder is a common symptom in aging and neuro- 52 degenerative disorder such as AD and dementia, as well as developing disorders such as 53 autism. Sleep disorders produce significant impairments in social and occupational func- 54 tions, which increase the social burden.” There are no references – that need to be corrected.

Response 3: As per your comments, we rewrote this part and added the references as follows.

“Sleep is essential for emotional, physical and cognitive functions. Sleep has restorative effects after activity, allowing optimal functioning [1, 2]. Sleep is responsible for maintaining brain homeostasis, and sleep disturbances including sleep deprivation and loss are closely associated with cognitive impairment, systemic inflammation and neurodegeneration [3, 4]. Therefore, sleep disturbances are potent risk factors and common symptoms for various neurological disorders, including neurovascular diseases, depression, attention deficit hyperactivity disorder (ADHD), autism spectrum disorder (ASD) and neurodegenerative diseases [5-8].”

Point 4. Describing the therapy for sleep disorders should be much more focused on disorders – the second most common sleep disorder is obstructive sleep apnea – completely different treatment options are used for this disorder. Further, putting benzodiazepines and phenobarbitals on the same scale as an antidepressant is misleading – the mechanisms of the medications are different, and the short-long term effect and need to increase the dose are not similar at all. This narrative seems to be designed to enhance the value of the studied therapeutic – which is again misleading for the reader.

Response 4: We appreciate your helpful comment. In this study, our experimental design and results are foused on sleep loss and poor sleep quality, and Maydis stigma has benefits in sleep disorder such as insomnia, but not obstructive sleep apnea. Obstructive sleep apnea has different treatment options such as positive airway pressure, weight control and mandibular advacement devices. As per your kind comments, we rewrote this point in introduction and dicussion part as follows:

Introduction)

Treatment of sleep disorders such as insomnia largely includes drug therapy and non-medical therapy. Medications such as benzodiazepines, barbiturates, anti-depressants and melatonin agonists are the easiest to use and fastest to act; however, there are concerns about adverse events arising from drug overuse, such as resistance with long-term use, reactions in the event of disruption, and withdrawal symptoms [13]. Non-pharmacological therapy for insomnia consists of cognitive behavioral therapies for insomnia (CBT-i), which is a type of therapy used for behavior and lifestyle changes, and complementary medicines including natural sleep aids and herbal remedies are used. Recently, interest in alternative medicines such as medicinal plants and dietary supplements is growing, although medication treatment has a fast and good effect for insomnia, there is a high potential for abuse and dependence [14]. 

Discussion)

Pharmacological treatments have been used for various types of sleep disorder such as insomnia. Current medications including benzodiazepines, barbiturates and melatonin agonist should be prescribed by a doctor due to extensive adverse effects, such as dependence and abuse [75]. BZDs and barbiturates are classified as Schedule IV controlled substances under the US Drug Enforcement Administration (DEA). BZDs are frequently prescribed to treat insomnia, because they reduce sleep latency and increase total sleep duration by enhancing the inhibitory action of GABA. However, BZDs have severe side effects such as rebound insomnia and anterograde amnesia. In addition, accumulation of active metabolites can cause confusion, cognitive dysfunction and depressive features [76] and can produce residual sleep medication effects including (e.g., drowsiness, difficulty concentrating, and impaired memory) and interfere with quality of life [77]. Ramelteon as a melatonin agonist, can be used to treat insomnia, and is the only non-scheduled prescription drug in the United States. Second line medications consist antidepressants and antihistamines have moderate level of tolerability. Some antidepressants used to treat insomnia are associated with increased suicidal ideation, increased mania/hypomania, and exacerbation of restless legs syndrome. Further, anticonvulsants can produce daytime sedation, dizziness and cognitive impairment. Residual sleep medication effects are particularly vulnerable to elderly individuals. Along with the increase of the elderly population, the number of people suffering from sleep disorders is also increasing. Therefore, there has been an increased interest to find a new approach to sleep problems with less adverse effects. Based on this reason, there is increasing interest in nutraceuticals, which is safe even if taken for a long time, for treatment of sleep disorders [14]. 

Point 5. There is no discussion of possible interaction of the used chemical – to induce sleep and alertness as well as the therapeutic – this should be widely discussed as a limitation or described to show a lack of such interactions.

Response 5: Thank you for the helpful comments. To explain the possible mechanism of MS on sleep improvement, we examined the expression of clock genes known to regulate sleep. we included a result on the expression of clock genes in the brain as figure 5, and also add this point in discussion part.

Result)

Figure 5. Effects of Maydis stigma extracts on the expression of clock genes in the brain. Mice administered MSE (10, 100 mg/kg) for 7 days. After sleeping test, the mice were sacrificed, and the brain was dissected. qRT-PCR was performed to measure the expression of clock genes including Per1 and 2, Cry1 and 2, Clock and Bmal1 in the hypothalamus. *P<0.05, **P<0.01; Veh vs MSE (N=3).

MS extracts increased the expression of circadian clock genes

Previous studies showed melatonin directly acts on the SCN regulating the circadian rhythms. A functional MT1 receptor is also required for the regulation of clock genes such as Bmal1 and Clock expression in the brain [33, 34]. The regulation of circadian rhythms is based on the transcriptional feedback loop of clock genes including PerCryClock, and Bmal1. To examine the molecular mechanism of MSE related to circadian clock genes, we performed the mRNA expression of circadian clock-regulated markers including PerCryClock, and Bmal1 using qRT-PCR. Our results showed that MSE (100 mg/kg) significantly increased the expression of clock genes including Per1/2Cry1/2 and Bmal1 in the hypothalamus (3.3/3.78, 2.07/2.37 and 2.46 folds, respectively) compared to vehicle control (Figure 5). These results demonstrated that MSE regulates the expression of circadian clock gene via activation of melatonin receptor, and thereby improves the sleep quantity and quality.”

Discussion)

Circadian rhythms in physiology and behavior are regulated by a central clock located within the SCN of the hypothalamus. The circadian clock is controlled by light and melatonin, regulates sleep and wake cycles, and has been shown to promotes alertness during the day [52]. The biological clock is based on the transcriptional/translational feedback loop of clock genes (PerCryClock, and Bmal1). von Gall et al. reported that Per1Cry1Clock, and Bmal1 were reduced in MT1 KO mice, but were expressed in WT and MT2 KO mice. Therefore, melatonin, acting through the MT1 receptor, is an important regulator of rhythmic clock gene expression in the pars tuberalis (PT) of the pituitary [33, 34]. Furthermore, melatonin treatment induces the clock gene to reverse erythroblastosis virus (REV-ERBα) expression in the rat hypothalamus [53]. In this study, our results showed that MSE increased the expression of clock genes including Per1/2, Cry1/2 and Bmal1 with the increase of melatonin receptors expression. von Gall et al.’s report [33] supports our findings that the increase of melatonin receptors expression is associated with the expression of clock genes. Therefore, these results suggest that MSE has a sleep promoting effect through an increase of the melatonin receptor and clock genes expression, and these effects may be caused by an increase in melatonin release. All of these results demonstrated that MSE has a novel effect on sleep improvement, and these effects may be associated with the activation of clock genes as well as melatonin receptors, and thereby can be a potential candidate for treating sleep disorders related with sleep loss and poor quality. However, further studies are needed to elucidate regulatory mechanisms of clock genes and melatonin receptors by MSE.

Point 6. Most of the discussion is just a description that should be in some part provided in the introduction. The discussion of the results compared to available data is very limited compared to the whole section.

Response 6: As per your comment, we rewrote this point in discussion section. We greatly appreciate the kind comments.

Reviewer 2 Report

This article title “Novel therapeutics for treating Sleep Disorders: new perspectives of Maydis stigma” by Kim et al., describes use of active ingredients derived from MSE and its effect on sleep, along with change in MT1/2 using qRT-PCR.  Kim et al., show that MSE extract decreased sleep latency and increased sleep duration, and Both MS extracts and beta-sitosterol increased the alpha activity in EEG analysis.

            This is an important study and has a potential in depicting role of Maydis stigma on sleep physiology and its therapeutic valuation.

However, I have some major concerns, which are described below. Addressing these issues should serve to strengthen this manuscript and increase confidence of readers in the conclusions.

Major Concerns

1.      Whole manuscript needs to be revised in terms of citing original articles. Author did not cite original article while describing sleep and its other biological function.  Manuscripts also need editing in terms of writing and presenting a coherent thought in abstract as well some major improvement in introduction.

a.      For Example, in line 41 author say “suppression of all body muscles”. Sleep doesn’t suppress body muscle, rather sleep suppress cholinergic muscular tone there by restricting muscle movement.

b.      Line 45: “Sleep interacts”: It is not meaningful to say Sleep interacts;

c.       Line 46 to 49: All functions has been referred by using only one reference, which is not correct.

d.      Missing reference for line 54-55

e.      A major flaw in writing: line 62: “Many medicinal plants have been a good source for various health problems of human.” Please correct the sentence here, I assume author mean by saying “Many medicinal plants have been a good source for treatment of various health problems of human.”

f.        Line 65-66: correct English here.

g.      Line 71; reference missing and not cited properly for Melatonin has various beneficial effects …..drug abuse. Please cite the original paper.

h.      Line 113: Provide rationale: “The lyophilized extracts are consecutively divided into Ethyl acetate, n-butanol and DW”.

i.        Correct English in line 181

j.        Correct English in line 278-279

2.      A second major concern is that it is unclear why only male mice were used as experimental subjects or and why experimental subjects were restricted to a single sex. I cannot find a statement about this anywhere in the manuscript. If only males were used this needs to be explicitly stated in the methods section and a justification needs to be provided. In that case, the potential for sex effects needs to be described in the discussion section. if both male and female mice were used, then this needs to be explicitly stated and, critically, sex effects should be assessed in the results section. If there were significant effects of sex how do the authors interpret this?

3.      Author must provide sample EEG traces in result section and show the differences observed between vehicle vs treatment.

4.      It is not clear if MS extract contain the B-sitosterol; Did author confirm using GC-MS or with any other method?

5.      There is major difference in saying quantification of mRNA vs receptor being synthesized from mRNA. In this study author did not quantify MT1/2 receptor protein level then why author use the term ‘receptor’. Author needs to be consistence in their use of term aligned with their finding.

Over all manuscript need a major revision in terms of data presentation and writing of results.

Author Response

First of all, we would like to thank you for handling our manuscript. We appreciate the important points raised by this comment. We believe this comment helped a lot to improve the quality of the manuscript. We added this point throughout the manuscript including introduction, method, result and discussion. We tried to answer the suggestions raised by reviewers as faithfully as possible. We believe that expert comments from the reviewers helped a lot to improve the quality of the present works.

Point 1: Whole manuscript needs to be revised in terms of citing original articles. Author did not cite original article while describing sleep and its other biological function.  Manuscripts also need editing in terms of writing and presenting a coherent thought in abstract as well some major improvement in introduction.

Response 1: As per your comment, we revised whole manuscript and updated the references. We greatly appreciate the kind comments.

1. For Example, in line 41 author say “suppression of all body muscles”. Sleep doesn’t suppress body muscle, rather sleep suppress cholinergic muscular tone there by restricting muscle movement.

Response: As per your careful comment, we corrected this sentence,

“Sleep is essential for emotional, physical and cognitive functions. Sleep has restorative effects after activity, allowing optimal functioning [1, 2].”

2. Line 45: “Sleep interacts”: It is not meaningful to say Sleep interacts;

Response: As per your comment, we revised this sentence and corrected it.

“Sleep is responsible for maintaining brain homeostasis, and sleep disturbances including sleep deprivation and loss are closely associated with cognitive impairment, systemic inflammation and neurodegeneration [3, 4].”

3. Line 46 to 49: All functions has been referred by using only one reference, which is not correct.

Response: As per your comment, we added the references.

“Therefore, sleep disturbances are potent risk factors and common symptoms for various neurological disorders, including neurovascular diseases, depression, attention deficit hyperactivity disorder (ADHD), autism spectrum disorder (ASD) and neurodegenerative diseases [5-8].”

4. Missing reference for line 54-55

Response: We added the references.

“Therefore, sleep disorders significantly impair social and occupational function, thereby increasing socioeconomic burden [11, 12].”

References)

  1. Streatfeild, J., et al., The social and economic cost of sleep disorders. Sleep, 2021. 44(11).
  2. Mattila, T., et al., Changes in the societal burden caused by sleep apnoea in Finland from 1996 to 2018: A national registry study. Lancet Reg Health Eur, 2022. 16: p. 100338.

5. A major flaw in writing: line 62: “Many medicinal plants have been a good source for various health problems of human.” Please correct the sentence here, I assume author mean by saying “Many medicinal plants have been a good source for treatment ofvarious health problems of human.”

Response: As your careful comment, we corrected this sentence as follows.

“Many medicinal plants including herbal remedies are good sources for treatment of various health problems in humans.”

6. Line 65-66: correct English here.

Response: As per your careful comment, we corrected this sentence as follows. Thank you for your careful comments.

“In recent years, with increasing interest in sleep induction and maintenance, new, effective, and safe sleeping pills have been developed.”

7. Line 71; reference missing and not cited properly for Melatonin has various beneficial effects …..drug abuse. Please cite the original paper.

Response: As per your comments, we added new citations and updated the references in this manuscript.

“Melatonin has various beneficial effects on sleep and circadian abnormalities, mood disorders, learning and memory, neuroprotection, drug abuse, and cancer by activating the high-affinity G protein-coupled receptors MT1 and MT2 [15-18].”

8. Line 113: Provide rationale: “The lyophilized extracts are consecutively divided into Ethyl acetate, n-butanol and DW”.

Response: We appreciate your helpful comment. We prepared the fractionation of Maydis stigma extracts according to previously reported information, and obtained it from EnsolBio Science Inc. of South Korea (http://ensolbio.co.kr/eng/index.html). We added this in method part.

“After extraction, Maydis Stigma extracts was concentrated at 80–90% and freeze-dried. The extraction yield was found to be 13%. Thereafter, the fractionation of lyophilized extracts was performed in the order of ethyl acetate, n-butanol, and DW (EnsolBio Sciences Inc.) [28, 29].”

9. Correct English in line 181

Response: As per your careful comment, we corrected this sentence as follows;

“A software package (Ethovision, Noldus IT BV, Wageningen, Netherlands) was used to record animal behavior.”

10. Correct English in line 278-279

Response: As per your comment, we corrected this sentence as follows. Thank you for careful comments.

“Interestingly, our data demonstrated that MSE treatment had a sleep-inducing and sleep-maintaining effect.”

Point 2: A second major concern is that it is unclear why only male mice were used as experimental subjects or and why experimental subjects were restricted to a single sex. I cannot find a statement about this anywhere in the manuscript. If only males were used this needs to be explicitly stated in the methods section and a justification needs to be provided. In that case, the potential for sex effects needs to be described in the discussion section. if both male and female mice were used, then this needs to be explicitly stated and, critically, sex effects should be assessed in the results section. If there were significant effects of sex how do the authors interpret this?

Response 2: We appreciate the helpful comment. We added this point in method section, and discussion section as follows.

Animals)

“Male ICR mice (20–30 g; four weeks old) were purchased from Orient Bio (Gyeonggi, Korea) and used for experiment.”

Discussion)

Most studies on sleep deprivation were either measured mainly in male subjects or that gender factors were not considered, although sleep deprivation is correlated with gender-specific differences and several clinical studies have shown sex difference in the insomnia and other sleep pathologies [40]. Koehl et al., suggested that there was no difference between genders in mice experiment on sleep deprivation [41]. However, female rodents, like humans, show changes in sleep and locomotor activity across their estrous cycle [42]. In addition, sleep studies in rodents showed that sex differences in sleep were eliminated by removal of sex hormones and gonadectomy. Therefore, in this study, we only used male mice as experimental subjects to avoid the effects of estrous cycle.

Point 3: Author must provide sample EEG traces in result section and show the differences observed between vehicle vs treatment.

Response 3: As per your comment, we added the data in result section as follows.

Figure 8. Effects of Maydis stigma on alpha waves (8-12 Hz) in mice. Mice administered Maydis stigma extracts (MSE; 20 mg/kg), b-sitosterol (BS; 10 mg/kg) and positive control (Diazepam, DZP, 2 mg/kg) for 7 days orally. Each bar represents the mean ± S.E.M. of the total power of alpha waves for 5 mins during the habituation period, before, and after treatment with pentobarbital sodium (i.p.). (A) Representative images of EEG traces during treatment, (B) Alpha activity by MSE administration, (C) Alpha activity by b-sitosterol administration. **p < 0.01, ****p < 0.0001 vs. Veh (N=7-8). Pre: pre: habituation; Treat: drug treatment; PB: pentobarbital sodium injection.

Point 4: It is not clear if MS extract contain the B-sitosterol; Did author confirm using GC-MS or with any other method?

Response 4: Unfortunately, we couldn’t do this analysis. But, we found vital information about Maydis stigma (corn silk) from the report on the raw materials and ingredients of Health Functional Foods of the Korean Food and Drug Administration (KFDA):compounds detected in corn silk included volatile oil (0.2%), alpha-terpineol, menthol, thymol, flavonoids, maysin, maysin-3-ehtylether, bitter substances, saponin (2-3%), tannins proanthocyanidins, sterols, beta-sitosterol, ergosterol, alkaloids (0.05%), 6-methoxybenzoxazolinone and fatty oil (2%), and beta-sitosterol uses as marker compound or biologically active compound. However, to develop it as a health functional food, we will need to submit analysis data on beta-sitosterol content as biologically active compound to KFDA.

Point 5: There is major difference in saying quantification of mRNA vs receptor being synthesized from mRNA. In this study author did not quantify MT1/2 receptor protein level then why author use the term ‘receptor’. Author needs to be consistence in their use of term aligned with their finding.

Response 5: As per your comment, we examined the protein expression of melatonin receptor1 and 2 in the frontal cortex of brain, and added the result as figure 3C.

Figure 3. Effects of Maydis stigma extracts on the expression of melatonin receptors 1 and 2 in the brain. Mice administrated MSE (1, 10, 100 mg/kg) and positive control (Valerian root extracts; VR, 10 mg/kg) for 7 days. After sleeping test, the mice were sacrificed, and the brain was dissected. (A) mRNA expression of MT1 and MT2 in the hypothalamus, (B) mRNA expression of MT1 and MT2 in the frontal cortex. *P<0.05, **P<0.01; Veh vs MSE (N=8-12). (C) Protein expression of MT1 and MT2 in the frontal cortex. *P<0.05, **P<0.01; Veh vs MSE (N=3).

Reviewer 3 Report

1.     As sleep is correlated with melatonin secretion, it’s very important to know the melatonin level after  Maydis Stigma treatment. I will recommend performing this experiment.

2.     It would be better to find out the molecular mechanism of Maydis Stigma with circadian clock-regulated markers including CLOCK, PER1, PER2, and BMAL1 because these markers are regulated by sleep.

3.     Western blot data makes me confused. Strongly recommended to add the raw file of western blot with full membrane.

Author Response

First of all, we would like to thank you for handling our manuscript. We appreciate the important points raised by this comment. We believe this comment helped a lot to improve the quality of the manuscript. We added this point throughout the manuscript including introduction, result and discussion. We tried to answer the suggestions raised by reviewers as faithfully as possible. We believe that expert comments from the reviewers helped a lot to improve the quality of the present works.

Point 1: As sleep is correlated with melatonin secretion, it’s very important to know the melatonin level after Maydis Stigma treatment. I will recommend performing this experiment.

Response 1: We appreciate your helpful comment. To measure the melatonin level, we need to collect the blood from experimental mice and Maydis stigma extracts. However, do this, we need a longer time (at least 4 weeks). Instead, we examined the protein expression of melatonin receptor 1 and 2 using Western blot, and we demonstrated that MS increased the protein expression of melatonin receptor 1 and 2 in the frontal cortex of brain, and added the result as figure 3C. I hope you can understand this.

Figure 3. Effects of Maydis stigma extracts on the expression of melatonin receptors 1 and 2 in the brain. Mice administrated MSE (1, 10, 100 mg/kg) and positive control (Valerian root extracts; VR, 10 mg/kg) for 7 days. After sleeping test, the mice were sacrificed, and the brain was dissected. (A) mRNA expression of MT1 and MT2 in the hypothalamus, (B) mRNA expression of MT1 and MT2 in the frontal cortex. *P<0.05, **P<0.01; Veh vs MSE (N=8-12). (C) Protein expression of MT1 and MT2 in the frontal cortex. *P<0.05, **P<0.01; Veh vs MSE (N=3).

Point 2: It would be better to find out the molecular mechanism of Maydis Stigma with circadian clock-regulated markers including CLOCK, PER1, PER2, and BMAL1 because these markers are regulated by sleep.

Response 2: As per your helpful comment, we add the result on the expression of clock genes in the brain as figure 5, and also added this point in the discussion part.

Result)

Figure 5. Effects of Maydis stigma extracts on the expression of clock genes in the brain. Mice administered MSE (10, 100 mg/kg) for 7 days. After sleeping test, the mice were sacrificed, and the brain was dissected. qRT-PCR was performed to measure the expression of clock genes including Per1 and 2, Cry1 and 2, Clock and Bmal1 in the hypothalamus. *P<0.05, **P<0.01; Veh vs MSE (N=3).

MS extracts increased the expression of circadian clock genes

Previous studies showed melatonin directly acts on the SCN regulating the circadian rhythms. A functional MT1 receptor is also required for the regulation of clock genes such as Bmal1 and Clock expression in the brain [35, 36]. The regulation of circadian rhythms is based on the transcriptional feedback loop of clock genes including PerCryClock, and Bmal1. To examine the molecular mechanism of MSE related to circadian clock genes, we performed the mRNA expression of circadian clock-regulated markers including PerCryClock, and Bmal1 using qRT-PCR. Our results showed that MSE (100 mg/kg) significantly increased the expression of clock genes including Per1/2Cry1/2 and Bmal1 in the hypothalamus (3.3/3.78, 2.07/2.37 and 2.46 folds, respectively) compared to vehicle control (Figure 5). These results demonstrated that MSE regulates the expression of circadian clock gene via activation of melatonin receptor, and thereby improves the sleep quantity and quality."

Discussion)

Circadian rhythms in physiology and behavior are regulated by a central clock located within the SCN of the hypothalamus. The circadian clock is controlled by light and melatonin, regulates sleep and wake cycles, and has been shown to promotes alertness during the day [56]. The biological clock is based on the transcriptional/translational feedback loop of clock genes (PerCryClock, and Bmal1). von Gall et al. reported that Per1Cry1Clock, and Bmal1 were reduced in MT1 KO mice, but were expressed in WT and MT2 KO mice. Therefore, melatonin, acting through the MT1 receptor, is an important regulator of rhythmic clock gene expression in the pars tuberalis (PT) of the pituitary [35, 36]. Furthermore, melatonin treatment induces the clock gene to reverse erythroblastosis virus (REV-ERBα) expression in the rat hypothalamus [57]. In this study, our results showed that MSE increased the expression of clock genes including Per1/2, Cry1/2 and Bmal1 with the increase of melatonin receptors expression. von Gall et al.’s report [35] supports our findings that the increase of melatonin receptors expression is associated with the expression of clock genes. Therefore, these results suggest that MSE has a sleep promoting effect through an increase of the melatonin receptor and clock genes expression, and these effects may be caused by an increase in melatonin release. All of these results demonstrated that MSE has a novel effect on sleep improvement, and these effects may be associated with the activation of clock genes as well as melatonin receptors, and thereby can be a potential candidate for treating sleep disorders related with sleep loss and poor quality. However, further studies are needed to elucidate regulatory mechanisms of clock genes and melatonin receptors by MSE.

Point 3: Western blot data makes me confused. Strongly recommended to add the raw file of western blot with full membrane.

Response 3: As per your comment, we added the raw file as follows. But these blots are not whole membrane, they are cut membrane. When we perform Western blot, blot was cut according to the molecular size of the target protein after transfer. And then blots are incubated with the primary antibody. Unfortunately, we cannot find the original blot of p-aCaMKII/-aCaMKII. So, we performed another experiment with remaining previous samples and the results were similar to the previous one. We are also attaching the mentioned new blots. I hope you can understand this.

Round 2

Reviewer 1 Report

The authors adequately responded to the comments and sufficiently improved the manuscript. 

Author Response

Dear sir, 

Thank you for your handing our manuscript. 

Best regards,

Kwon Kyoung Ja

Reviewer 2 Report

Over all manuscripts has improved  a lot and I thank authors on working on suggestions. 

Author Response

Dear sir,

Thank you for handling our manuscript. 

Best regards.

Kwon Kyoung Ja

Reviewer 3 Report

Authors do not address the two most important points properly. 

Point 1: As sleep is correlated with melatonin secretion, it’s very important to know the melatonin level after Maydis Stigma treatment. I will recommend performing this experiment.

Point 3: Western blot data makes me confused. Strongly recommended to add the raw file of western blot with full membrane.

Author Response

First of all, we would like to thank you for handling our manuscript. We believe this comment helped a lot to improve the quality of the manuscript. We added this point in result section. We tried to answer the suggestions raised by reviewers as faithfully as possible. We appreciate the important points raised by this comment.

Point 1: As sleep is correlated with melatonin secretion, it’s very important to know the melatonin level after Maydis Stigma treatment. I will recommend performing this experiment.

 Response 1: As per your comment, we measured the melatonin level after performing the new experiment. We added the result on the melatonin levels in plasma of animals as figure 3D as follows. We appreciate your helpful comment.

Method)

Measurement of melatonin levels

After sleeping test, blood samples were collected from the heart under deep anesthesia. After centrifugation at 1,000 g at 4 °C for 10 min, separated plasma was withdrawn and stored at -80 °C. Melatonin levels were measured using an enzyme-linked immunosorbent assay (ELISA) kit (Abcam, Cambridge, UK) according to the manufacturer’s protocol.

Results)

“To evaluate whether MSE administration increased the melatonin secretion, melatonin levels were measured in the blood of mice after MSE administration. Compared with the vehicle control, the plasma melatonin level is significantly increased in the MSE administration group (~1.36 folds). The results indicate that MSE can effectively increase melatonin levels.”

Figure 3. Effects of Maydis stigma extracts on the expression of melatonin receptors 1 and 2 in the brain. Mice administrated MSE (1, 10, 100 mg/kg) and positive control (Valerian root extracts; VR, 10 mg/kg) for 7 days. After sleeping test, blood from the mice was collected, and the brain was dissected. (A) mRNA expression of MT1 and MT2 in the hypothalamus, (B) mRNA expression of MT1 and MT2 in the frontal cortex. *P<0.05, **P<0.01; Veh vs MSE (N=8-12). (C) Protein expression of MT1 and MT2 in the frontal cortex. (D) melatonin levels from plasma. *P<0.05, **P<0.01, ***P<0.001; Veh vs MSE (N=3).

Point 3: Western blot data makes me confused. Strongly recommended to add the raw file of western blot with full membrane.

Response 3: As per your comment, we added the raw file as follows. Previously, when we performed Western blot, blot was cut according to the molecular size of the target protein after transfer. And then blots are incubated with the primary antibody. Unfortunately, we did not provide the full blots. So, we performed another experiment with new samples and the results were similar to the previous one. We are attaching the new blots for reviewer. I hope you can understand this.

Thank you in advance for your consideration of this manuscript.

I look forward to hearing from you.

Sincerely yours,

Kyoung Ja Kwon, PhD

Round 3

Reviewer 3 Report

1. Need to check the primer sequence (18S rRNA)

2. There are copious controversies about beta-sitosterol against psychiatric disorders. So it would be better to rewrite the abstract and conclusion.

3. Check the sentence pattern and grammatical errors.

Author Response

We would like to thank you for handling our manuscript. We carefully revised the manuscript and we tried to answer the suggestions raised by reviewers as faithfully as possible.

Point 1: Need to check the primer sequence (18S rRNA)

Response 1: We appreciate your helpful comment. We corrected 18S rRNA sequence as follows.

18S rRNA:

Sense-CATTAAATCAGTTATGGTTCCTTTGG

Antisense-TCGGCATGTATTAGCTCTAGAATTACC

Point 2. There are copious controversies about beta-sitosterol against psychiatric disorders. So it would be better to rewrite the abstract and conclusion.

Response 2: As per your comments, we rewrite this point as follows.

Abstract

- β-Sitosterol (BS) is a phytosterol and a natural micronutrient in higher plants, and has a similar structure to cholesterol. It is a major component of MS and has anti-inflammatory, anti-depressive, and sedative effects.

- Our results demonstrate for the first time that MS has a sleep-promoting effect via melatonin receptor expression, which may provide new scientific evidences for its use as potential therapeutic agents for the treatment and prevention of sleep disturbance.

Conclusion

Collectively, our results suggest that Maydis stigma are safe and potential nutraceuticals for various psychiatric disorders associated with sleep disorders without adverse effects such as addiction and toxicity.

Point 3. Check the sentence pattern and grammatical errors.

Response 3: We carefully revised the manuscript. Thank you for your careful comment.

Thank you in advance for your consideration of this manuscript.

I look forward to hearing from you.

Sincerely yours,

Kyoung Ja Kwon, PhD
